# Manipulation via Membranes: High-Resolution and Highly Deformable Tactile Sensing and Control

**Miquel Oller**    **Mireia Planas**    **Dmitry Berenson**    **Nima Fazeli**

Department of Robotics, University of Michigan
Ann Arbor, MI 48109, United States
{oller, mireiap, dmitryb, nfz}@umich.edu
https://www.mmintlab.com/manipulation-via-membranes

**Abstract:** Collocated tactile sensing is a fundamental enabling technology for dexterous manipulation. However, deformable sensors introduce complex dynamics between the robot, grasped object, and environment that must be considered for fine manipulation. Here, we propose a method to learn soft tactile sensor membrane dynamics that accounts for sensor deformations caused by the physical interaction between the grasped object and environment. Our method combines the perceived 3D geometry of the membrane with proprioceptive reaction wrenches to predict future deformations conditioned on robot action. Grasped object poses are recovered from membrane geometry and reaction wrenches, decoupling interaction dynamics from the tactile observation model. We benchmark our approach on two real-world contact-rich tasks: drawing with a grasped marker and in-hand pivoting. Our results suggest that explicitly modeling membrane dynamics achieves better task performance and generalization to unseen objects than baselines.

**Keywords:** Manipulation, tactile control, deformable tactile sensors

## 1    Introduction

Tactile sensing collocated at the contact interface between the robot and environment is a key enabling technology for dexterous manipulation [1]. The resulting information-rich and highly discriminative tactile cues are far more informative than proprioceptive external joint torques. To date, a number of high-resolution collocated tactile sensors have gained traction in the robotics community including Soft Bubbles [2], GelSlim [3], DIGIT [4], and FingerVision [5]. However, there are two coupled chal-

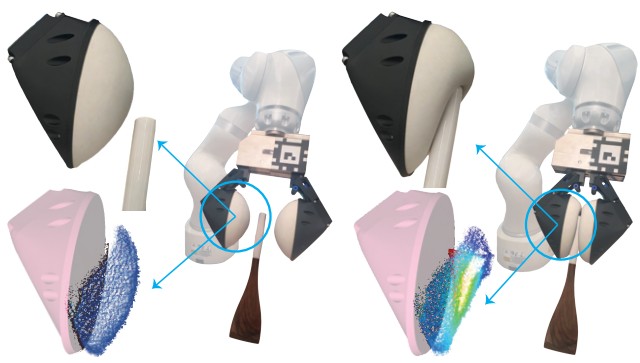

Undeformed Membrane          Deformed Membrane

Figure 1: **Membrane Deformation Visualization** The sensor membranes deform significantly as a result of their interaction with the grasped object and environment. (top) real deformation, (bottom) perceived deformation.

lenges in using this class of sensors: the high-dimensionality of the sensor signal (signature) and the sensor dynamics introduced between the robot and what it is contacting.

Recent progress in exploiting these high-dimensional signatures has resulted in sensor specific algorithms for state-estimation [6, 7, 8, 9, 10, 11] and controls [12, 13, 14, 15]. However, most progress has addressed low-deformation sensors with negligible dynamics (e.g., GelSlim, DIGIT, and FingerVision). These dynamics cannot be ignored for high-deformation tactile sensors (e.g., Soft Bubbles). This is because large sensor deformations lead to both non-negligible relative motion of the grasped object with respect to the end-effector and incipient slip as the object is brought into contact with the environment. When used effectively, this high compliance is desirable for contact rich interactions as it allows for large and stable contact patches as well as gradual force build up.

6th Conference on Robot Learning (CoRL 2022), Auckland, New Zealand.

In this paper, we first illustrate deformation and force transmission differences between hard and soft tactile sensors by showing the relative behavior of the Soft Bubbles and GelSlim 3.0. The mechanical interface of the former is an inflated latex membrane while the latter uses a dense polymer. This exercise motivates our main contribution: a method to learn soft tactile sensor membrane dynamics that accounts for the sensor deformations caused by the physical interaction between the grasped object and environment. Our approach integrates the perceived 3D geometry of the sensor membrane with proprioceptive reaction wrenches and predicts future membrane deformations conditioned on robot actions. A key feature of our method is the decoupling of the sensor membrane dynamics from the tactile observation model. Tactile observation models compute features such as the object pose and extrinsic contact location from the sensor state (3D geometry and reaction wrenches). This decoupling exploits the fact that sensor membrane dynamics are shareable across tasks because they are inherent to the sensor mechanics while the observation model can be task-specific. We demonstrate how our method enables reasoning over the joint dynamics of the robot, grasped object, and environment to enable precise control of object pose and force transmission and empirically compare our method against 5 baselines on two real-world contact rich robotic tasks.

## 2    Related Work

There are two major types of tactile sensing: localized and distributed. Localized sensing, here referring to the use of a single force-torque sensor often at the robot wrist, summarizes external contact information as a wrench composed by six numbers, 3 linear and 3 torque terms. Localized sensing can be provided by the robot through joint torques or using F/T sensors [16, 17, 18]. In contrast to localized, distributed tactile sensing collocated at the contact interface can provide information dense feedback in the form of images [2, 3, 5] or pressure distributions [19]. The survey by Yamaguchi and Atkeson [20] summarizes recent advances in these sensors. With the promise of information dense sensing; however, comes challenges including novel dynamics at the contact interface, and tactile signature representation for state-estimation, prediction, and controls.

Finding good tactile sensing representations is challenging because it typically requires either expert knowledge or considerable amounts of data. Approaches such as TACTO [21] and Narang et al. [11] have reduced data requirements by simulating DIGIT [4] and SynTouch Biotac pressure-based sensors [19], respectively. Building on this, Kelestemur et al. [22] propose an object pose estimator trained in simulation that transfers to the real-world. However, current tactile simulators only support a small set of relatively rigid sensors and simulation of soft sensor, much like other deformables, remains a significant challenge. Therefore, our data is collected from real-world interactions.

Early work exploited dense tactile signals in closed-loop controllers for contact servoing [12] or active exploration [13]. More recently, tactile control approaches have been used in robotic bi-manual object manipulation [23], deformable object manipulation [24], and peg-in-hole insertion [25]. These later approaches use the GelSlim tactile sensor [3] which exhibits small and approximately linear deformations. This simplified model is not effective for the Soft Bubbles and assuming sticking external contact is restrictive in many applications (see Sec. 4.1 and 5).

Closely related to our work, previous work has explored explicitly modelling the tactile sensor dynamics. Van Hoof et al. [15] projected the Biotak tactile signals into a learned low-dimensional representation where dynamics are linear. Other approaches such Tian et al. [14] have modelled the dynamics of a vision-based tactile sensor such as GelSight. Their entire control pipeline is formulated in the sensor space, requiring the goal states to be specified as tactile states. Instead, we decouple the sensor dynamics from the task dynamics, allowing us to specify tasks in terms of object poses and forces transmitted to the environment. Lambeta et al. [4] extended their control formulation to a multi-finger control setting by adding structure to the tactile dynamics predictions. This approach conditions the dynamics on the estimated object pose, which creates a dependency on the pose estimation and may limit the model's ability to generalize to different objects.

## 3    Problem Formulation

The goal of the robot is to control the grasped object pose and the force transmitted to the environment as the object is brought into contact. Our central idea is to represent these "task dynamics" by

modeling them using the tactile sensor membrane dynamics. We are motivated by the fact that the grasped object pose and transmitted forces are resolved simultaneously with the membrane deformation and end-effector reaction wrenches. This means the robot can plan for desired task states by reasoning through the membrane dynamics – i.e., planning actions that result in membrane deformations that in turn result in the desired poses and transmitted forces.

Let $\boldsymbol{p}_t$ denote the 3D pose of any points on the surface of the membrane, then the dynamics model $\boldsymbol{p}_{t+1} = f(\boldsymbol{p}_t, \boldsymbol{w}_t, \boldsymbol{a}_t, \boldsymbol{z}_t)$ represents how each point deforms conditioned on the current membrane geometry, end-effector reaction force $\boldsymbol{w}_t$, robot action $\boldsymbol{a}_t$, and the grasped object $\boldsymbol{z}_t$. Next, let $\boldsymbol{q}_t$ denote the object-environment configuration, then the observation model $\boldsymbol{q}_t = g(\boldsymbol{p}_t, \boldsymbol{w}_t, \boldsymbol{z}_t)$ maps the current membrane geometry and reaction force to configurations. We define the task state as $\boldsymbol{x}_t = (\boldsymbol{q}_t, \boldsymbol{w}_t)$ and goal as $\boldsymbol{x}_g$. Planning and control reduces to solving the optimization problem:

$$\min_{\boldsymbol{a}_t} \quad \sum_{t=0}^{N} (\boldsymbol{x}_t - \boldsymbol{x}_g)^T \mathbf{Q} (\boldsymbol{x}_t - \boldsymbol{x}_g) + \boldsymbol{a}_t^T \mathbf{R} \boldsymbol{a}_t$$

$$\text{s.t.} \quad \boldsymbol{p}_{t+1}, \boldsymbol{w}_{t+1} = f(\boldsymbol{p}_t, \boldsymbol{w}_t, \boldsymbol{a}_t, \boldsymbol{z}_t), \quad \boldsymbol{q}_t = g(\boldsymbol{p}_t, \boldsymbol{w}_t, \boldsymbol{z}_t)$$

Figure 2: $\boldsymbol{p}_t$: membrane state, $\boldsymbol{w}_t$: external wrench, $\boldsymbol{z}_t$: object geometry, $\boldsymbol{a}_t$: robot action.

where $\mathbf{Q}$ and $\mathbf{R}$ are positive definite matrices weighing the relative importance of reaching goal states with effort. The key to solving this problem is learning a sufficiently-accurate estimate of $f$, which is the focus of this paper.

## 4 Methods

In this section, we first illustrate characteristic deformation and force transmission differences between soft and hard sensors (Soft Bubbles vs. GelSlim 3.0) to motivate the need for our membrane dynamics. Next, we present our main contribution: our approach to learning a soft tactile sensor membrane dynamics model that accounts for the sensor deformations caused by the physical interaction between the grasped object and environment as well as an observation model used to extract the task state. Finally, we describe the controller which leverages these dynamics for task execution.

### 4.1 Sensor Deformation Evaluation

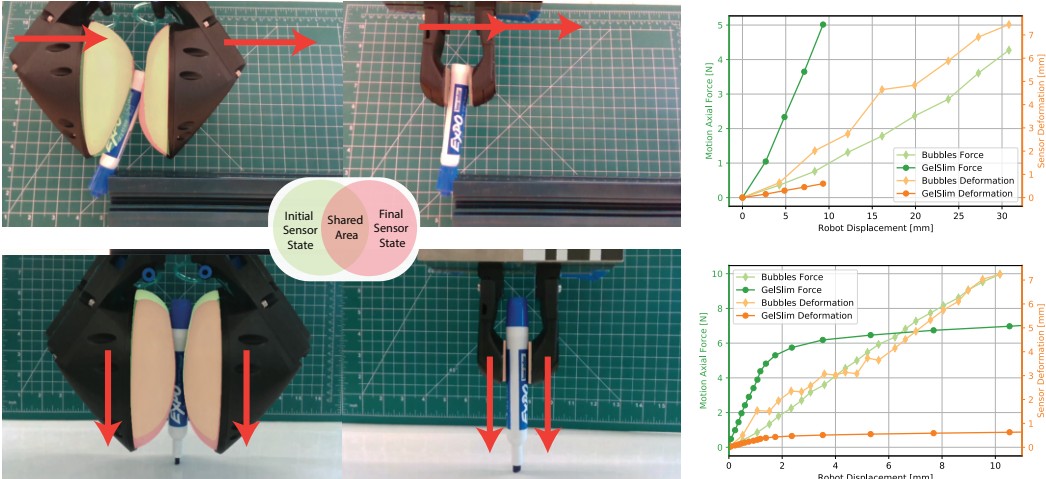

Figure 3: **Sensor Deformation Comparison** We compare the Soft Bubbles (left) and the GelSlim 3.0 (center) sensors on force transmission and deformation for two in-contact motions. Top row compares the sensors on an axial motion perpendicular to the grasp plane. Bottom row compares the sensor on a top-down axial motion in the grasp plane. Green overlay shows the sensor state before contact. Red overlay shows the deformed sensor state as a result of the interaction.

Controlling both in-hand object pose and transmitted force is key for dexterous manipulation. In this section, we ask just how significant is the relative object displacement w.r.t. to the end-effector for soft vs hard sensors and how is transmitted force affected? Fig. 3 shows an illustrative comparison of the Soft Bubbles and Gelslim 3.0 for two in-contact motions along the sensor's main axes. The results show that the Soft Bubble membrane deforms an order of magnitude more than the GelSlim. We also observe that the object orientation varies significantly more (approx. $25°$) during tangential motion. Moreover, we observe significant slip between the object and GelSlim, while the Bubbles maintain sticking contact due to large contact patches resulting from the significant deformation.

The GelSlim deformations are less than 1 mm and the resulting force transmission profile is relatively sharp, similar to rigid-body interactions. We also observe force plateauing and relative slip much earlier in the vertical contact task. In contrast, the soft sensors' large compliance allows for a more gradual force transmission without slip for a larger range of motion. While this compliance is desirable for many contact-rich tasks, it must be accounted for during fine manipulation.

The large contact patches result in increased surface area which provides two useful features: increased perception of the object shape at contact and larger distribution of friction through a concave contact surface. The former point can improve in-hand object pose estimation because it provides more features for inference, and the latter can improve grasp stability owing to larger and more distributed frictional forces where the membrane "hugs" the object. The gradual build up of force is effectively a form of passive compliance. This compliance prevents damage to position controlled robots by mitigating rigid-on-rigid body contact. Further, it eases the burden on impedance control by providing local to contact compliance that can be perceived and controlled through robot motion. Thus, the remainder of this paper investigates how to learn the membrane dynamics of Soft Bubble sensors in order to realize the above benefits for dexterous manipulation.

## 4.2 Membrane Dynamics Model

The goal of the membrane dynamics model is to predict the tactile sensor membrane deformations $p_{t+1}$ and reaction forces $w_{t+1}$ as a function of their current geometry $p_t$, reaction force $w_t$, the robot action $a_t$, and the tool geometry $z_t$. The model has access to the current membrane geometry measured by a time-of-flight depth sensor mounted within the gripper. This membrane geometry $p$ can be represented as either a depth map image or a 3D point cloud using the camera intrinsics. Here, we use the depth map encoding as it is a more structured and simplifies pixel-wise correspondences.

The dynamics model predicts the pixel-wise deformation of the sensor membrane and the reaction force sensed at the end-effector using the architecture depicted in Fig. 4. In order to enable multi-step planning, the model also predicts the grasp frame pose reached by the robot $r_{t+1}$ as a consequence of taking the action $a_t$ (accounting for the robot impedance). Similar to [15], the model

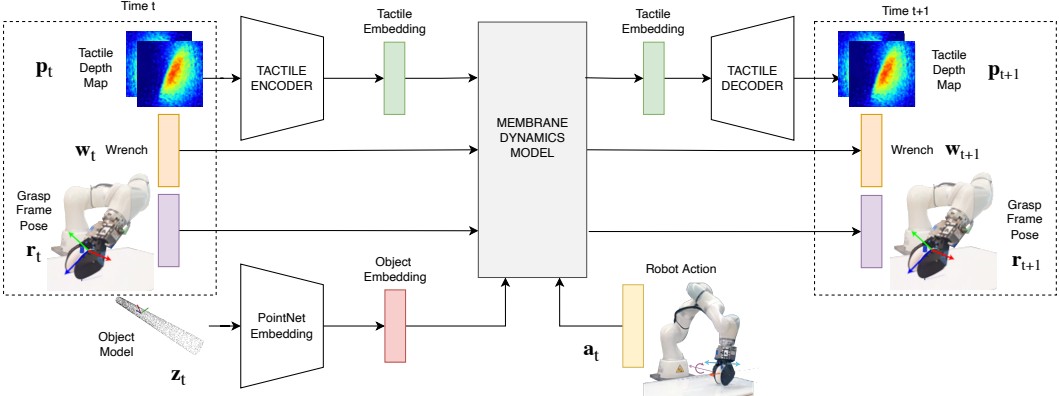

Figure 4: **Membrane Dynamics Model** Our proposed dynamics model predicts the membrane states $p$, external wrenches $w$ and grasp frame poses $r$ conditioned on the robot action $a$ and the grasped object model $z$. The high-dimensional tactile depth maps and object models are projected into a learned lower-dimensional space for more efficient processing. Tactile embedding projections are learned in an autoencoder fashion. Object models are encoded using a PointNet-based network.

encodes tactile images into a lower-dimensional latent representation. Intuitively, the model learns to exploit structure in the deformed membrane geometry to obtain a more compact representation. The object geometry is also encoded into a lower-dimensional representation to help the membrane dynamics model combine these multi-modal inputs. Since the object geometry is provided as a point cloud, our model uses a PointNet [26] inspired network pre-trained on ModelNet40 [27] and fine-tuned on our task data to obtain a lower-dimensional latent representation of the grasped object.

We train our dynamics model in two-steps. First, we learn the tactile image latent representation in an Autoencoder [28] fashion with only the tactile images from our training dataset. Our model is able to exploit tactile data from different tasks by training the tactile encoder-decoder on the combined dataset, subsequently sharing this latent tactile representation across tasks. We also pre-train the object geometry model embedding on ModelNet40 and freeze all weights but the last layer. Second, we freeze the tactile encoder-decoder and train the dynamics model end-to-end.

We train the dynamics model $f$ with data composed by state-action-state transitions conditioned on the object geometry embedding $(s_t, z_t, a_t, s_{t+1})$. In our case $s_t = (p_t, w_t, r_t)$. We employ a supervised reconstruction loss $\mathcal{L}_{\text{dyn}}$ on the predicted state composed of 3 mean squared error (MSE) terms, one for each of the three sensory modalities our model predicts: tactile, wrenches, and poses. The losses are aggregated based on weights $\alpha_i$ to compensate for the discrepancies in units and scale of each sensory contribution (see Appendix A.2 for implementation details).

$$\mathcal{L}_{\text{dyn}}(\hat{s}_t, s_t) = \alpha_1 \text{MSE}(\hat{p}_{t+1}, p_{t+1}) + \alpha_2 \text{MSE}(\hat{w}_{t+1}, w_{t+1}) + \alpha_3 \text{MSE}(\hat{r}_{t+1}, r_{t+1})$$
$$\text{where} \quad \hat{p}_{t+1}, \hat{w}_{t+1}, \hat{r}_{t+1} = f(p_t, w_t, r_t, z_t, a_t)$$

## 4.3 Observation Model

The observation model $g$ maps the current membrane geometry $p_t$ and end-effector reaction wrench $w_t$ to an estimate of the current grasped object pose $q_t$. It has access to the object geometry $z_t$, robot proprioception $r_t$, and the location of the environment contact surface. We decouple pose estimation from membrane dynamics, enabling flexibility in observation model choice and improving generalization across objects and tasks. Fig. 5a illustrates the observation model. First, we use the camera intrinsics and extrinsics to project the depth maps into the end-effector frame. Next, we extract contact points by comparing the current tactile images to their undeformed reference states. We extract contact points by computing pixel-wise deformation and selecting the top 10% that deform at least 3 mm. See Appendix A.3 for more details. Next, we use Iterative Closest Points (ICP) to estimate the object pose from the extracted contact points. When the grasped object is in contact with the environment, we project the predicted object pose to the object-environment contact manifold [29].

## 4.4 Controller

The goal of the controller is to drive the task states $x_t = (q_t, w_t)$, composed by object-environment configurations $q$ and transmitted forces $w$, to their goal values $x_g = (q_g, w_g)$ by solving the optimization problem posed in Sec. 3. To this

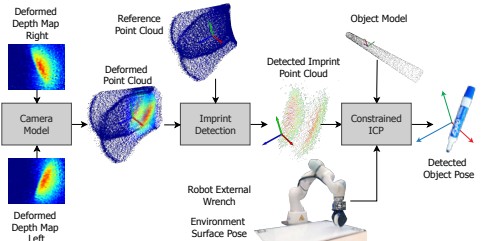

(a) **Observation Model** The deformed depth maps are projected using the camera intrinsics and extrinsics to obtain the joint pointcloud of the deformed sensors. Then this is compared with a reference pointcloud of the undeformed sensor state to extract the contact points. Finally, ICP fits an object model to the detected contact points to estimates the corresponding object pose.

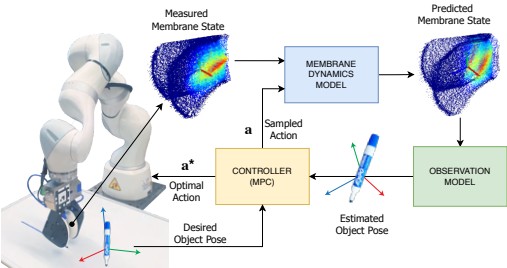

(b) **Control Pipeline** Given a measured state, our controller queries the membrane dynamics model with sampled actions to obtain the predicted membrane states. The object pose is estimated from the predicted membrane states and it is compared with the desired poses to compute the costs associated to the sample actions. The costs are aggregated and the resultant optimal action is executed by the robot.

Figure 5

end, we use model-predictive control; specifically, the Model-Predictive Path Integral (MPPI) controller [30]. MPPI is effective in handling continuous action spaces and can be parallelizable for an efficient implementation with neural network models. Fig. 5b illustrates the control pipeline. The controller iteratively optimizes a nominal control sequence by sampling action sequences and rolling out their dynamics using our learned dynamics model and observation model. These trajectories are sequences of task states $x_t, \ldots, x_{t+N}$, and they are obtained by first using our learned dynamics model to obtain dynamic states $s_i$, and then using the observation model to obtain the correspondent task states $x_i$. Finally, the trajectory costs are computed by comparing the predicted task states, with the desired values under a quadratic cost and the nominal control sequence is updated. For pivoting, the goal object orientations are defined as the desired tool orientation with respect to the robot. For drawing, the goal object orientations are perpendicular to the whiteboard along the drawing outline. In both cases the desired forces are perpendicular to the environment surface to encourage contact. See Appendix A.4 for implementation details.

## 5 Experiments and Results

We demonstrate our proposed approach to tactile control in two real-world tasks: drawing with a grasped marker and in-hand pivoting of objects. Both of these tasks require the robot to apply forces through the grasped object to the environment while controlling its pose with respect to the end-effector. Since the grasp is not rigid, the object may move w.r.t. the end-effector due to deformations or slip. Effective task execution must account for relative motion and prevent failure due to the tool slipping out of the grasp.

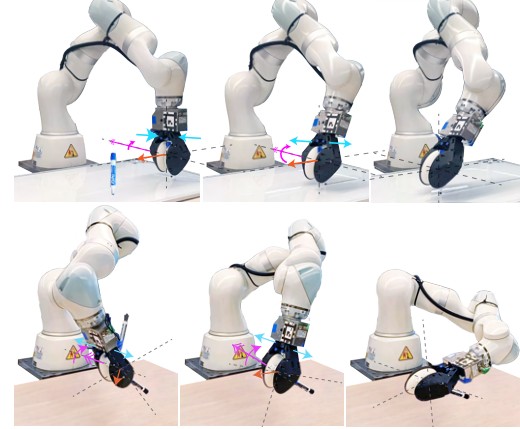

Figure 6: **Action Spaces** The action spaces for drawing (top) and pivoting (bottom) are composed of the grasp width, displacement along the grasp plane, and rotation about axis perpendicular to this plane.

Our method uses the same dynamics model architecture (Fig. 4) and learned latent membrane geometry representation across tasks. One key benefit of learning membrane deformations as opposed to object poses is that although each task has different dynamics, the membrane deformation dynamics are inherent to the sensor. This allows us to combine data from multiple tasks to learn the membrane latent representation. However, since each task has particular dynamics and action spaces, the membrane dynamics function (Fig. 4 gray block) is task-specific.

We evaluate our two tasks on multiple tools, 8 for drawing and 8 for pivoting, Fig. 7. For each task, we split our tools into two sets: we collect training data with 5 tools (train object set), and use the remaining 3 to test how well our learned model generalizes to unseen objects (test object set). Using the 5 training tools, we collect 800 state-action-state triplets per tool, obtaining a total of 4000 samples to train the models. Our data collection combines random samples with epsilon-greedy samples from a Jacobian controller (see Appendix B.1 for details).

### 5.1 Baselines

We benchmark our proposed dynamics model against 5 baselines inspired by related approaches. The original baselines only consider tactile signatures as inputs. Hence, we extend each to also incorporate reaction wrenches, robot poses, and object geometries. We also modify the baselines such that all evaluated methods use the same observation model. This way, all methods have access to the same information for a fair comparison. Our first baseline, *Bubble Linear Dynamics* is inspired by [15] where the dynamics are constrained to be linear in the tactile latent space. Our second baseline, *Object Pose Dynamics*, is inspired by [4] where the object in-hand pose dynamics are modeled as opposed to membrane dynamics. We adapt this baseline by replacing the object pose estimator with our observation model for consistency across all methods. Our third baseline, *Fixed Model*, is only used in the drawing task. It assumes that there are no dynamics and that the membrane state remains constant. This baseline illustrates the importance of modeling membrane deformations. Our

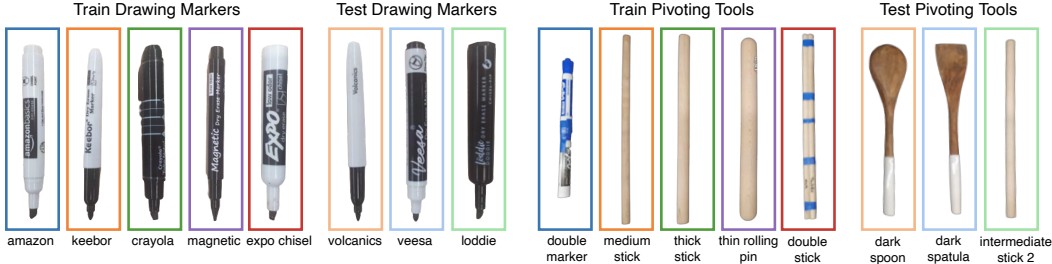

Figure 7: **Evaluated Tools** (left) drawing object set, (right) tools used for pivoting.

fourth baseline, *Jacobian*, is only used in the pivoting task and is inspired by [12]. This baseline assumes that the grasped object is fixed w.r.t. the environment during contact. Our final baseline does not use a dynamics model. Instead, the controller takes pseudo-random actions. For drawing, pseudo-random actions move the robot along the drawing path while uniformly randomly selecting the end-effector orientation, grasp width, and height from the action space. For pivoting, these actions bring the object into contact with the table while randomly selecting subsequent end-effector motion and grasp width. This baseline evaluates the complexity of each task.

## 5.2    Drawing with a Marker

The goal of this task is to control the marker pose and force transmitted through its tip to draw a desired shape while preventing the marker from slipping out of the grasp. For simplicity, we choose a continuous 0.6 meter line. Fig. 6 shows the robot action space composed of Cartesian displacement in the plane of the line, rotation about an axis perpendicular to this plane, and the gripper width.

We evaluate the drawing performance using a camera to extract a binary mask of the white board and comparing it to the desired binary image (see Appendix B.2 for more details). The drawing score is a value between 0 and 1, representing the percent of the line the robot has successfully drawn. The goal states provided to the controller are a sequence of poses where the marker tip is perpendicular to and in contact with the board along the desired drawing path. Fig. 8 shows the drawing score distributions. Table 1 summarizes the drawing evaluation scores. We perform 10 trials per marker. The trials end when the final desired pose is reached, if the maker slips out of the hand, or if the robot can no longer correct the marker pose because of joint limits or collisions. Our results show that nonlinear membrane model (ours) outperforms the baselines both in training and test objects. For train objects, our model successfully draws on average a 10.0% more of the desired line than the second best baseline (*Bubble Linear Dynamics*), and a significant 32.6% more than *Object Pose Dynamics*. For test objects, the difference is 7.4% and 25.8% respectively. We believe this is due to better robustness to uncertainty, generalization, and expressivity (see Sec. 6). Our results also show that *Object Pose Dynamics* performs fairly similar to *Fixed Model*. Moreover, in general test objects perform slightly better on average than the training objects. A possible explanation is that their geometry and tip properties make the test objects easier to control, something we could not know *a priori*. The "magnetic" brand marker is particularly challenging likely due to its fine tip. Removing this marker from the evaluation metrics eliminates the discrepancy between test and training sets, suggesting that the model has effectively learned important salient features of the task that it can indeed generalize to novel markers.

## 5.3    In-Hand Pivoting

The goal of this task is to drive the grasped object to a desired in-hand configuration solely using the environment to modify its configuration w.r.t the end-effector. Fig. 6 shows the action space of the pivoting task which is composed of Cartesian motion in the plane of the gripper jaws, rotation about an axis perpendicular to this plane, and the gripper width. Our metric for pivoting performance is error in desired vs. realized tool orientation. We use our observation model to estimate the final pose. The desired tool pose is sampled at random within the graspable range. The controller optimizes the pivoting action sequence to achieve the final pose by exploiting the dynamics model. Table 1 summarizes the pivoting evaluation scores (see Appendix C.1 for the evaluation distributions). We perform 10 pivoting executions per tool. The trials end when reaching the final desired pose

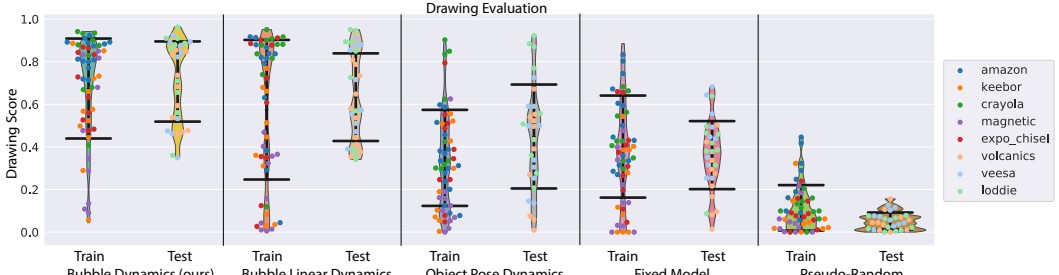

Figure 8: **Drawing Evaluation Results** In pairs, we show the train (left) and test (right) scores for each of the evaluated methods. Colored dots indicate each tool achieved score (10 per tool). Black horizontal lines indicate the sample standard deviation around the mean. The drawing score represents the percent of the desired drawing successfully drawn, being 1.0 the optimal.

within, or after having performed 10 pivoting actions. If the tool slips out of hand, we record the last achieved angle. Our results show that our proposed model outperforms the baselines in reaching desired in-hand orientations, obtaining the lowest orientation error on average. Our method is also the most consistent, obtaining the lowest standard deviation. We highlight that our method obtains an orientation error as low as half the second best approach. Our results also show our method is less prone to overshooting errors compared to baselines.

| Representation | Drawing Scores | | | | Pivoting Scores [deg] | | | |
| | Train Objects | | Test Objects | | Train Objects | | Test Objects | |
| | Mean ↑ | Std ↓ | Mean ↑ | Std ↓ | Mean↓ | Std↓ | Mean↓ | Std ↓ |
|---|---|---|---|---|---|---|---|---|
| Bubble Dynamics | **0.674** | 0.235 | **0.707** | 0.188 | **6.96** | 12.2 | **5.41** | 6.33 |
| Bubble Linear Dyn. | 0.574 | 0.328 | 0.633 | 0.206 | 14.4 | 19.8 | 11.5 | 12.8 |
| Object Dynamics | 0.348 | 0.225 | 0.449 | 0.244 | 24.2 | 38.9 | 16.8 | 26.9 |
| Fixed / Jacobian | 0.402 | 0.239 | 0.361 | 0.159 | 22.8 | 36.7 | 9.39 | 20.5 |
| Pseudo-Random | 0.114 | 0.107 | 0.050 | 0.040 | 16.9 | 16.9 | 24.6 | 37.4 |

Table 1: **Evaluation & and Task Score Statistics:** Drawing scores = percentage of the drawing the robot has successfully completed represented as a number between 0 and 1 (higher = better). Pivoting scores = absolute value of the error in desired vs goal orientation. (lower = better). Statistics are reported over 10 trials per representation.

## 6 Discussion and Limitations

In this paper, our experiments show that decoupling membrane dynamics from object dynamics and characterizing the sensor dynamics improves task performance as well as the ability to use novel unseen tools. We believe that modelling sensor dynamics performs better than object dynamics because: i) sensor dynamics explicitly considers the combined compliance of the robot and the sensor, ii) are more robust against observation noise, and iii) can share tactile signatures across tasks and objects. Our observation model has some stochasticity, and its variance can highly impact trajectories that roll out from its observations. Instead, our approach rolls out dynamic trajectories agnostic to the observation model. As a result, the final pose prediction is less effected by the observation variability. See Appendix A.6 for a detailed comparison.

A limitation of our approach is that it assumes rigid objects with known geometry. This assumption simplifies the observation model, planning, and controls, since it restricts object-environment interactions to rigid on rigid. While the membrane dynamics model may not be affected significantly by the object compliance, our method will require a more sophisticated observation and controls framework. Another limitation of our model is the quality of long-horizon predictions. Long-horizon predictions can be poor because recursive model calls may lead to out of distribution predictions. This is the main reason why we use a planning horizon of 2 steps. Some methods to mitigate this effect are multi-step prediction methods [31] or flow-based projections [32]. Another direction to explore is developing new membrane representations that are robust to this effect. Finally, our model predictions are smooth. As a consequence, our method may fail to model drastic state changes like the outcomes of hitting obstacles, or dropping the tool. This could be addressed by leveraging the structure of non-smooth/hybrid dynamics [33] during representation learning.

## Acknowledgments

This research project is supported by Toyota Research Institute under the University Research Program (URP) 2.0. This work has been partially supported by the mobility grants program of Centre de Formació Interdisciplinària Superior (CFIS) - Universitat Politècnica de Catalunya (UPC). We would also like to show our gratitude to the anonymous reviewers for their helpful comments in reviewing the paper. We also thank the members of the Manipulation and Machine Intelligence (MMINT) Lab for their support and feedback.

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
