# OpenReview forum: "Manipulation via Membranes: High-Resolution and Highly Deformable Tactile Sensing and Control"
_robot-learning.org/CoRL/2022/Conference — CoRL 2022 Poster_

### Official Review · Reviewer_rK46 · 2022-07-26

**Originality:** Very Good
**Technical Quality:** Very Good
**Clarity Of Presentation:** Very Good
**Impact:** 4

**Recommendation:**

Weak Accept: I recommend accepting the paper, but will not argue for my recommendation if the majority of other reviewers have a different opinion.

**Summary:**

The paper proposes a method to learn soft tactile sensor membrane dynamics that accounts for sensor deformations caused by the physical interaction between the grasped object and environment.  Authors explore the differences between soft(large deformation) and hard sensors, and demostrate two manipulation tasks with a Soft Bubbles.

**Issues:**

 no explicit issues.

**Quality Of The Limitations Section:**

Limitations are addressed clearly

**Reviewer Expertise:**

4: The reviewer is confident but not absolutely certain that the evaluation is correct

**Robotics Focus:**

Sufficient demonstration on hardware

**Strengths And Weaknesses:**

Strengths:
1.  explore the differences between soft and hard sensors.
2. propose a method to learn the large deformation sensor dynamics.
3. benchmark the method on two real-world contact-rich tasks.

Weaknesses:
I do not see explicit weakness in the paper.



**Summary Of Recommendation:**

In manipulation tasks, tactile sensation plays an important role. Comparing to hard sensor, this paper uses a large deformation sensor to collecting rich contact information, and complete contact-rich tasks.
The membranes seem like a physical filter, may smooth the object geometry, and filter the details of the objects. How it performs in a fine manipulation tasks, such as recognize two sides of usb flash drive or manipulating a small object?

---

> ### Author Response · Authors · 2022-08-28
> **Response to Reviewer rK46 -- Summary of our revisions, additions, and rebuttals**
>
> First of all, we would like to thank you very much for your time and effort invested in reviewing our work. We highly value your diligence during the reviewing process and are happy to learn that you have found our work highly original, clear, and of major impact.
>
> We also appreciate the reviewer’s insightful comments about the membranes and how they may act as a physical filter. These insights demonstrate the reviewer’s deep understanding of tactile sensors. We do notice the smoothing effect of the membranes compared to other hard sensors such as the GelSlims. However, this effect is not impactful for the types of manipulation tasks we study in this paper. As you mentioned, high-resolution features can be useful for tasks such as object detection or pose estimation when the object has fine features that can be used to disambiguate effectively. We are interested in investigating the effects of this smoothing in future studies.
>
> Thank you again for your feedback and comments. We have made a number of improvements to the manuscript including extending the Appendix to include more implementation details. We hope that you find the additions and edits satisfactory and we’d be very grateful if you were to consider elevating your recommendation.

---

### Official Review · Reviewer_ke9K · 2022-07-28

**Originality:** Very Good
**Technical Quality:** Fair
**Clarity Of Presentation:** Poor
**Impact:** 4

**Recommendation:**

Weak Accept: I recommend accepting the paper, but will not argue for my recommendation if the majority of other reviewers have a different opinion.

**Summary:**

I found the paper somewhat opaque, but I think the aim is to model the state of the tactile sensor as a collection of poses on the contact surface, then use model-predictive control (i.e. rollout action trajectories to a goal state) to achieve desired states of the contact surface, and in so doing manipulate held tools (constrained to long thin tools that can be held and pivoted by the gripper). This capability was demonstrated by drawing a line with various shaped markers and pivoting objects to a desired angle.

**Issues:**

Please see comments above.

**Quality Of The Limitations Section:**

Additional details required

**Reviewer Expertise:**

2: The reviewer is willing to defend the evaluation, but it is quite likely that the reviewer did not understand central parts of the paper

**Robotics Focus:**

Sufficient demonstration on hardware

**Strengths And Weaknesses:**

Strengths
- Interesting combination of tasks. I think the focus towards tool use (which this paper is basically about) is a good direction to take the field.
- The use of model-predictive control on tactile data is a step forward in the field.

Weaknesses
- I did find the paper difficult to decode. The paper was generally well written, but perhaps it is coming from a different viewpoint from how I see things. I was even mystified on the first sentence, when I was wondering what 'collocated' meant (and whether it has an extra 'l'...).  New terminology seems to be dropped in willy-nilly, e.g. saying there is localized and distributed tactile sensing (isn't what they call localized tactile sensing just force-torque sensing, which is why 'force and tactile' are usually grouped together?) This was all dropped in without any clear problem statement of what the paper is contributing.
- I welcome the attempt to compare two distinct tactile sensors (Soft Bubble and GelSlim), but then this just stopped after the first results section 4.1. What is the point of having this comparison if you're not going to use it in the rest of the paper? If throughout the paper you'd compared both sensors (even if one of them does not work well on the tasks) I would have seen the study as a lot stronger. As it was, I was first looking for results with the GelSlim, then wondering if there are none what are the concrete criticisms of the sensor that makes it not worth pursuing (and if so why not just delete the GelSlim from the paper entirely?)
- I could not understand how your controller works. How are you setting the goal? How does that relate to the deformation of the membrane? How is that set? How does this relate to the tactile data have from the sensor? This was all unclear to me.

**Summary Of Recommendation:**

I think this is fundamentally interesting and novel work, but I find the presentation so opaque that I cannot judge for sure.

---

> ### Author Response · Authors · 2022-08-28
> **Response to Reviewer ke9K -- Summary of our revisions, additions, and rebuttals -- Part 1**
>
> We would like to thank you very much for the time and effort invested in reviewing our work. We highly value your diligence during the reviewing process and assessing the quality of our work. We appreciate that you regard our work as original and of major impact.
>
> Regarding the summary statement you provided, it's very close with one small difference: Our approach models the entire membrane and not just the contact surface (Referencing Section 3 – Problem Statement and Section 4.2 – Membrane Dynamics Model). The observation model does indeed use the contact surface of the membrane to compute the pose of the object; however, this component is independent of the membrane model.
>
> Please find below our responses, edits, and additions to the specific weaknesses you highlighted:
>
> **Weakness 1: I did find the paper difficult to decode. The paper was generally well written, but perhaps it is coming from a different viewpoint from how I see things. I was even mystified on the first sentence, when I was wondering what 'collocated' meant (and whether it has an extra 'l'...). New terminology seems to be dropped in willy-nilly, e.g. saying there is localized and distributed tactile sensing (isn't what they call localized tactile sensing just force-torque sensing, which is why 'force and tactile' are usually grouped together?) This was all dropped in without any clear problem statement of what the paper is contributing.**
>
> We have focused our efforts on making our work more clear and accessible and rely on prior work for the usage of terminology. Regarding the specific example you mention (“collocated” – Section 1, line 20 ) we do provide a reference to its usage [1]. Collocated tactile sensing is an established term (e.g., see [2] for a review paper by Chris Atkeson and [3] for their usage in a recent publication) used to refer to sensing at the contact interface. The term distributed tactile sensing is also established, for several examples, see [4,5,6] below. We do agree that localized is somewhat ambiguous and have modified the text to “Localized sensing, here referring to the use of a single force-torque sensor often at the robot wrist, summarizes external contact information as a wrench composed of six numbers, 3 linear and 3 torque terms.” We chose localized vs distributed sensing to emphasize that the focus is on spatial distribution and location as opposed to the type of sensor feedback (force, deformation, proprioception, etc.).
>
> The final comment regarding a lack of a problem statement or clear contribution, we refer to Line 43 through 54 and Section 3 – Problem Formulation for contributions and problem statements.
>
> [1] A. Rodriguez. The unstable queen: Uncertainty, mechanics, and tactile feedback.Science Robotics, 6 (54):eabi4667, 2021.
>
> [2] Yamaguchi, A., & Atkeson, C. G. (2019). Recent progress in tactile sensing and sensors for robotic manipulation: can we turn tactile sensing into vision?. Advanced Robotics, 33(14), 661-673.
>
> [3] Chaudhury, A. N., Man, T., Yuan, W., & Atkeson, C. G. (2022). Using Collocated Vision and Tactile Sensors for Visual Servoing and Localization. IEEE Robotics and Automation Letters, 7(2), 3427-3434.
>
> [4] Tsujiuchi N, Koizumi T, Ito A, et al. Slip detection with distributed-type tactile sensor. 2004 IEEE/RSJ International Conference on Intelligent Robots and Systems (IROS). Vol. 1. 2004. p. 331–336.
>
> [5] Hosoda K, Tada Y, Asada M. Anthropomorphic robotic soft fingertip with randomly distributed receptors. Rob Auton Syst. 2006;54(2):104–109.
>
> [6] Tomo TP, Schmitz A, Wong WK, et al. Covering a robot fingertip with uskin: a soft electronic skin with distributed 3-axis force sensitive elements for robot hands. IEEE Robot Automat Lett. 2018;3(1):124–131.
>
> [7] Manuelli, L., & Tedrake, R. (2016, October). Localizing external contact using proprioceptive sensors: The contact particle filter. In 2016 IEEE/RSJ International Conference on Intelligent Robots and Systems (IROS) (pp. 5062-5069). IEEE.

---

> > ### Author Response · Authors · 2022-08-28
> > **Response to Reviewer ke9K -- Summary of our revisions, additions, and rebuttals -- Part 2**
> >
> > Continuing from part 1, we review the remaining weaknesses.
> >
> > **Weakness 2: I welcome the attempt to compare two distinct tactile sensors (Soft Bubble and GelSlim), but then this just stopped after the first results section 4.1. What is the point of having this comparison if you're not going to use it in the rest of the paper? If throughout the paper you'd compared both sensors (even if one of them does not work well on the tasks) I would have seen the study as a lot stronger. As it was, I was first looking for results with the GelSlim, then wondering if there are none what are the concrete criticisms of the sensor that makes it not worth pursuing (and if so why not just delete the GelSlim from the paper entirely?)**
> >
> > The primary focus of our work was to develop dynamic models, observation models, and controller formulations to enable dexterous manipulation with the Soft Bubble tactile sensors. The purpose of the illustrative comparison is to motivate the need to explicitly consider sensor deformations (in the case of the soft sensor) and provides context to show how distinct the hard and soft sensors are. This is particularly pertinent as the GelSlim’s have gained significant traction while the Bubbles may not be all too familiar to most researchers.
> > There are two main challenges in comparing tactile sensors across tasks: Metrics and Task Subjectivity. Metrics: we currently do not have established and widely accepted metrics to compare tactile sensor performance in a meaningful way – this itself would be a very interesting research question, though one that is very difficult to definitively answer. For example, the level of sensor resolution needed to achieve two tasks can be wildly different, shear estimation may not be a requirement for some tasks, and task performance may mask the specifics of why a sensor performs better in one task and how to modify designs such that this metric can be maximized. This latter issue leads to Task Subjectivity, meaning depending on the task (and potentially objects used in the task), one sensor could perform better than the other, even slight tweaks to tasks could flip results. While it may be interesting to have compared the sensors, it is unclear what conclusions could be drawn given the size of the study – we emphasize that there is no particular criticism of the GelSlim’s that would prohibit such a comparison; however, there would also be no clear conclusion to be drawn.
> > In response to why not remove this section entirely, as the other reviewers note, it is helpful to have context for the sensor and motivation for the technical contributions of the paper.
> >
> > **Weakness 3: I could not understand how your controller works. How are you setting the goal? How does that relate to the deformation of the membrane? How is that set? How does this relate to the tactile data have from the sensor? This was all unclear to me.**
> >
> > Thank you for highlighting the need for further details and clarifications. To address this, we have added several sections to our manuscript, specifically:
> > * Explain more clearly how our controller implementation works (Section 4.4 Line 202 through 210) – this section highlights how the goals are set and how they relate to the membrane deformation. Briefly, the goals are described as desired object poses and external forces transmitted to the environment. For example, in-hand pivoting goals are specified as the desired pose of the object in the grasp visualized as the green cylinder in the video. The observation model (Section 4.3 and Fig. 5a) is responsible for computing the object pose given membrane geometry, proprioception, and F/T at the wrist. The membrane states evolve according to the membrane dynamics model, and the membrane states are related to the goals through the observation model. The controller chooses actions (end-effector motions) to minimize the cost (distance to desired object poses and external forces).
> > * Added additional implementation details and how goal task states are specified (Section 4.4 and Appendix A.4) – this section expands on the goal specification with a formal cost function and constraints as well as implementation details in software. The Appendix A.4 also contains detailed cost and constraints.
> > * Added performance metrics for the controller computations (Appendix A.4)
> >
> > The information added, in addition to the code that we will release and maintain, will allow other researchers to implement our controller in their domains.
> >
> > Thank you again for your feedback and comments. We have made a number of improvements to the manuscript including extending the Appendix to include more implementation details. Our revisions are marked in blue for easy reference. We hope that you find the additions and edits satisfactory.

---

### Official Review · Reviewer_gpA9 · 2022-07-30

**Originality:** Good
**Technical Quality:** Good
**Clarity Of Presentation:** Very Good
**Impact:** 3

**Recommendation:**

Weak Accept: I recommend accepting the paper, but will not argue for my recommendation if the majority of other reviewers have a different opinion.

**Summary:**

The paper proposed a membrane dynamic model of the soft Bubble sensor. The model takes the perceived 3D geometry of the membrane, wrench of manipulator’s end-effector, robot actions, and object geometry to predict future membrane deformations and wrench. A key feature of the model is that the decoupling of the sensor dynamics from the task dynamics so that the membrane dynamic model may generalize to different objects and tasks.

**Issues:**

1.	Considering the complexity of the controller, it is of interest to know what is the computational time cost of updating the optimal action, which is an important characteristics for controller?
2.	The authors are suggested to also add experiment of drawing standard circles, which should be more convincing than the simple straight line.
3.	In section 5.2, the experimental results show that the test objects perform even slightly better than the training objects, which is possibly because the test objects are easier to control according to the authors. However, this makes the conclusion on the generalization of the proposed model somehow questionable?


**Quality Of The Limitations Section:**

Limitations are addressed clearly

**Reviewer Expertise:**

4: The reviewer is confident but not absolutely certain that the evaluation is correct

**Robotics Focus:**

Sufficient demonstration on hardware

**Strengths And Weaknesses:**

The main strengths of the paper are:
1.	The problem formulation and methods are well presented in words and in mathematical notations that can be easily understood. The supplementary materials are also well organized and include enough implementation details of the learning process.
2.	The main feature of the proposed model is stated clearly and is emphasized and explained in a couple of places, which is helpful for the readers.

The main weaknesses of the paper are:
1.	The membrane dynamic model is relatively complex combining a variety of sensory data and known information on the object’s geometry, which limits the usage of the proposed model.
2.	The setup of the two real-world tasks are relatively easy compared to what the soft membrane sensor is capable of as described in the paper. The authors might increase the difficulty of the tasks to some degree.


**Summary Of Recommendation:**

The paper is well written, and the proposed methods are clearly presented and supported by the hardware experimental results. The paper could be improved by addressing the weaknesses and issues. Hence, I recommend weak accept for the paper.

---

> ### Author Response · Authors · 2022-08-28
> **Reviewer gpA9 -- Summary of our revisions, additions, and rebuttals -- Part 1**
>
> We would like to thank you very much for your time and effort reviewing our work with such diligence. We’re happy to hear that you found our work easy to understand, well organized, and thoroughly detailed. Your suggestions and recommendations are highly valuable and in light of the wonderful feedback we received from you and our other reviewers, we have made a number of changes and additions that we believe have improved the quality of our manuscript. Specifically regarding the weaknesses you have highlighted:
>
> - **Weakness 1. The membrane dynamic model is relatively complex combining a variety of sensory data and known information on the object’s geometry, which limits the usage of the proposed model.**
>
>     The model we developed was the simplest architecture we could come up with that met the criteria of handling all the sensing modalities (tactile signature in the form of point clouds/depth maps and wrenches), object representation (PointNet embedding), and the robot action as well as to allow sharing tactile signatures across both tasks (using the tactile encoder-decoders shown in Figure 4). The inputs to the model have significantly different dimensions and distributions, requiring the proposed level of complexity. In terms of computational complexity, model evaluation is a forward pass of the neural network and runs at several tens of hertz. This speed is particularly highlighted during control where many evaluations of the model are required to compute trajectory rollouts. We have added computation speed as part of the implementation details of the controller in Appendix A.4.
>
> - **Weakness 2. The setup of the two real-world tasks are relatively easy compared to what the soft membrane sensor is capable of as described in the paper. The authors might increase the difficulty of the tasks to some degree.**
>
>     We appreciate the reviewer’s recommendation to increase the complexity of the tasks and their suggestion to attempt to draw a more complex geometry such as circles. However, despite appearing simple, these tasks are quite complex for the following reasons:
>
>     - The grasped object can move with respect to the robot gripper due to the sensor compliance. As the grasped object is brought into contact with the environment, the external force pushes the object and effectively “loads” the membrane, similar to how one would load a spring by compressing it. This effect is more complex for the membranes as they are a distributed and nonlinear element. Addressing the complexity introduced by these dynamics is central to our approach and model.
>
>    - In addition to the relative motion, the object can slip out of grasp or reach a configuration that can no longer be compensated by the robot. Maintaining a stable grasp is a challenge that our approach addresses.
>
>     - The action space is high-dimensional, composed of end-effector motion (6 DOF) and the gripper width (1 DOF). This high-dimensionality allows for more freedom in control at the cost of more variables to reason over.
>
>     - Variability in the object's initial pose – the robot is handed the tool at the beginning of each run, and as such, the initial pose is different from run to run.
>
>     - For the drawing task, an uncontrolled approach such as our pseudo-random can only achieve on average a 10% of the desired line.
>
>     Nonetheless, we hope that in the future we can extend the drawing task to draw more complex shapes as the reviewer has suggested.
>
> This concludes part 1 of our response, the following comment will continue part 2.

---

> > ### Author Response · Authors · 2022-08-28
> > **Reviewer gpA9 -- Summary of our revisions, additions, and rebuttals -- Part 2**
> >
> > Continuing from part 1, we now review the issues raised:
> >
> > - *Issue 1: Considering the complexity of the controller, it is of interest to know what is the computational time cost of updating the optimal action, which is an important characteristics for controller?*
> >
> >     Thank you for highlighting the need for further details and clarifications. To address this, we have added several sections to our manuscript, specifically:
> >
> >     - Explain more clearly how our controller implementation works (Section 4.4)
> >     - Add additional implementation details (Section 4.4 and Appendix A.4)
> >     - Add performance metrics for the controller computations (Appendix A.4). Table 4 specifically describes the computational cost broken down into components (model evaluation, pose estimation, and cost computation).
> >     - Explain how goal task states are specified. (Section 4.4 and Appendix A.4)
> >
> >     Currently, our main bottlenecks for computational speed are the observation model to estimate object pose and cost computation. To address these limitations in the future, we will work on incorporating more learning into the observation model to cache repeated computations and simplify the cost computation by either reducing the task horizon or simplifying the cost itself. The new details of the Appendix, in addition to the code that we will release and maintain, will allow other researchers to implement our controller in their domains.
> >
> > - *Issue 2: The authors are suggested to also add experiment of drawing standard circles, which should be more convincing than the simple straight line.*
> >
> >     We refer the reviewer to our response to Weakness 2 for experiment complexity. Regarding drawing circles, we simply did not have enough time to draw more complex shapes in the short rebuttal phase.
> >
> > - *Issue 3: In section 5.2, the experimental results show that the test objects perform even slightly better than the training objects, which is possibly because the test objects are easier to control according to the authors. However, this makes the conclusion on the generalization of the proposed model somehow questionable?*
> >
> >     As you have noted, there is a modest improvement in performance for the test tools over the training tools. For the drawing task, the culprit for the discrepancy between the test and training tool performances is the magnetic marker – it proved to be a very difficult tool to manipulate despite being in the training set. If its performance is removed from the metrics, then the discrepancy is also eliminated. This suggests that the model has effectively learned important salient features of the task that it can indeed generalize to novel objects. We’ve updated Line 276 to clarify this point. If the training results were worse for all tools, then there would certainly be a question on generalizability and model expressivity; however, the discrepancies are due to a single object in each task and the discrepancies are eliminated if the results from those specific objects are omitted. As such, we believe in the generalizability of the method.
> >
> > We thank you again for your feedback as it has led to significantly improved clarity and quality of the manuscript. Our edits and additions are marked in blue for easy reference. We hope that our responses, edits, and additions have satisfactorily addressed your concerns.

---

### Official Review · Reviewer_xvAZ · 2022-08-01

**Originality:** Good
**Technical Quality:** Very Good
**Clarity Of Presentation:** Very Good
**Impact:** 3

**Recommendation:**

Weak Accept: I recommend accepting the paper, but will not argue for my recommendation if the majority of other reviewers have a different opinion.

**Summary:**

The main contribution of the paper is a framework to learn a membrane dynamics model for their custom bubble membrane fingertips for use in tasks such as continuously pressing down on a whiteboard with a marker to draw a line and reorienting an object between the fingers by exploiting the environment (a table). There is a time of flight depth sensor inside of the bubble gripper that can generate a pointcloud of the surface of the gripper. With the depth information, they can use PointNet and other encoders and decoders to generate embeddings of objects that the bubble gripper is grasping. They can then learn the membrane dynamics model that can then be used to complete the 2 above tasks.

**Issues:**

I don't really have any issues with the paper. There were plenty of experiments and the results were presented in a satisfactory method. The paper was also well written and fairly straightforward. Thus, I don't really have any concerns with the paper.

**Quality Of The Limitations Section:**

Limitations are addressed clearly

**Reviewer Expertise:**

4: The reviewer is confident but not absolutely certain that the evaluation is correct

**Robotics Focus:**

Sufficient demonstration on hardware

**Strengths And Weaknesses:**

I really liked how the authors compared the sensor deformation between 2 sensors they had available. It helps to provide some context between the size and compliance of multiple sensors instead of only reporting the results of one sensors. I do know that this Bubble gripper has already been introduced in RoboSoft so this is not a hardware paper.
The limitations of the method and bubble gripper were also adequately raised in the paper and video so I don't have any concerns.
I also did like the overlay of the desired orientation of the object in the video as I was at first confused as to what the green line was.
If I were to nitpick something, I would have wanted to see if by using the gelsight slim and a 2D image embedding network, how would the performance compare with the PointNet based network. In addition, I would be curious to know how much data would be needed for each training to have adequate performance or equivalent results.


**Summary Of Recommendation:**

Overall, I think this paper is useful for understanding how to best use the depth information that the time of flight sensor inside the bubble gripper can be used to help the robot reorient an object for better tool use. I think it is valuable, although not many sensors are similar to the bubble gripper that provide a point cloud and would use PointNet. I feel like for other sensors like GelSight and Fingervision, having the images would be easier to handle than the point clouds, but that's just a difference in sensor types (2D vs 3D vision) so I can't fault it. I thought that the experiments were thorough with a variety of different objects of the similar cylindrical handle used. However, I would have liked if there had been an experiment of using the bubble gripper to use a screwdriver to tighten a screw like was teased in the video, but maybe its just due to the compliance of the bubble gripper that it would be unable to grip it strongly enough in order to produce the torque necessary to tighten. Anyways, I decided on the weak accept because I think that although there is a contribution in the paper, it seems like the method may only be applicable to this sensor unless there comes along a different sensor that also relies on point clouds.

---

> ### Author Response · Authors · 2022-08-28
> **Response to Reviewer xvAZ -- Summary of our revisions, additions, and rebuttals**
>
> We would like to thank you very much for the time and effort you invested in diligently reviewing our work. We also appreciate that you had taken the time to look into the supplementary materials as well, particularly because this section is almost as long as the paper itself. We’re happy to learn that you found our work clear, technically convincing, and well organized.
>
> We’ve made a number of additions and edits to help improve the paper. Specifically regarding your comments:
>
> - **How much data is required to get adequate performance (Sample Complexity):** With reference to the paragraph starting on line 230 and Appendix B.1 of the paper, our method required 800 state-action-state triplets per training object (5 training objects per task = 4000 total samples) to train the model such that the tasks could be achieved. The main time bottleneck is not in data collection (see Appendix B), rather it is in the evaluation of the model + controller. This is because for each object, we run 10 trials and there are 8 objects per task.
>
> - **Soft and Hard sensor Comparison:** We value the reviewer’s suggestions about comparing GelSlim performance across the proposed task adapting the model we propose. We did not compare the sensors in our proposed tasks since this was not the scope of our paper. The goal of our work is to enable novel functionality for the Soft Bubble sensors. The illustrative comparison we show motivates the need to explicitly consider sensor deformations and provides context to show how distinct the sensors are.
>
>     There are two main challenges in comparing tactile sensors across tasks: Metrics and Task Subjectivity. Metrics: we currently do not have established and widely accepted metrics to compare tactile sensor performance in a meaningful way – this itself would be a very interesting research question, though one that is very difficult to definitively answer. For example, the level of sensor resolution needed to achieve two tasks can be wildly different, shear estimation may not be a requirement for some tasks, and task performance may mask the specifics of why a sensor performs better in one task and how to modify designs such that this metric can be maximized. This latter issue leads to Task Subjectivity, meaning depending on the task (and potentially objects used in the task), one sensor could perform better than the other, even slight tweaks to tasks could flip results.
>
>     Nonetheless, we are definitely interested in trying this in future work. We expect some transferability of our model across sensors when adapting the Bubbles’ depth maps to the Gelslims’ RGB images, but a complete investigation is needed to verify our expectations.
>
> - **Extension to Other Tasks such as Screwing:** We show the screwing task illustratively in the video; however, as you correctly hypothesized, the sensors are not able to generate enough torque yet to reliably unscrew a tight screw. Further, to do a practical screwing task, we’d need to use a relatively sophisticated perception system to first localize the screw before engaging with it which was outside the scope of this project. Our approach is purely tactile and tracking the screw state would have required a more complex perception system. Nonetheless, we hope that in the future we can extend our system to be able to handle such tasks.
>
> Our revisions are marked in blue for easy reference. Finally, we would like to thank the reviewer again and we hope that our discussion and modifications address all issues.

---

### Meta-Review · Area_Chair_EExq · 2022-08-15

**Recommendation:** Accept (Poster)
**Confidence:** 4

**Metareview:**

Summary:
The focus of the paper is to learn a membrane dynamic model for a custom ‘bubble’ sensor.  Tasks include drawing a line on whiteboard with a marker, and pivoting objects using the fingers and a table.  A ToF sensor generates a point cloud of the internal membrane wall and allows for the use of learning techniques on it.  Soft and hard sensors are compared.
The contribution is highly original, and overall technical contributions are sound.  The paper is generally clear and well written and has the potential to have high impact in the research community. It is well aligned to CoRL and has a clear robotics use case.

Strengths:
The paper is easy to understand and backed up with informative supplementary materials.
The tasks are well-conceived, although perhaps a little simple, and have applicability for real-world use cases
The differences between hard and soft sensors is a good addition.
Limitations are clearly identified and discussed

Weaknesses
The proposed model is relatively complex, simpler models are not investigated but might provide a better trade off in terms of performance vs compute.
It would be great to get some idea on the complexity of the controller, and a clearer description of how it is set up and how it works.
Motivation must be stronger and clearer
The comparison between hard and soft sensors could have been carried on throughout the experimental section


***Update***
The authors were responsive to reviewer recommendation, and reviewer decisions were similar across the board.  I agree with the reviewers.